# A Qualitative Study on the Impact of First Steps—A Peer-led Educational Intervention for People Newly Diagnosed with Parkinson’s Disease

**DOI:** 10.3390/bs9100107

**Published:** 2019-10-10

**Authors:** Andrew Soundy, Johnny Collett, Sophie Lawrie, Shelly Coe, Helen Roberts, Michele Hu, Sally Bromley, Peter Harling, Alex Reed, Jan Coeberg, Camille Carroll, Helen Dawes

**Affiliations:** 1School of Sport, Exercise and Rehabilitation, University of Birmingham, Birmingham B15 2TT, UK; 2Centre for Movement, Occupational and Rehabilitation Sciences, Faculty of Health and Life Sciences, Oxford Brookes University, Oxford OX3 OBP, UK; jcollett@brookes.ac.uk (J.C.); 19003090@brookes.ac.uk (S.L.); scoe@brookes.ac.uk (S.C.); hdawes@brookes.ac.uk (H.D.); 3Academic Geriatric Medicine, University of Southampton. Southampton General Hospital Mailpoint 807, Southampton SO16 6YD, UK; H.C.Roberts@soton.ac.uk; 4Department of Neurology, Nuffield Department of Clinical Neurosciences, Level 3, West Wing, John Radcliffe Hospital, Oxford OX3 9DU, UK; michele.hu@ndcn.ox.ac.uk; 5Parkinson’s UK Oxford Branch, Botley Women’s Institute Hall, North Hinksey Lane, Oxford OX2 0LT, UK; oxford.sally@gmail.com; 6Norton Consulting Group, Malthouse, Main Road, Curbridge OX29 7NT, UK; peter.harling@nortonexecutive.com; 7European Parkinson’s Therapy Centre. Piazzale Delle Terme, 3, 25041 Dafo Boario Terme (Brescia), Italy; alexreeditaly@gmail.com; 8Department of Neurology, St George’s University Hospitals NHS Foundation Trust, London SW17 0QT, UK; Jan.Coebergh@asph.nhs.uk; 9Institute of Translational and Stratified Medicine, University of Plymouth, N14, ITTC Building, Plymouth Science Park, Plymouth Science Park, Plymouth, Devon PL6 8BX, UK; camille.carroll@plymouth.ac.uk

**Keywords:** Parkinson’s disease, therapy, storytelling, rehabilitation, illness narratives

## Abstract

*Aim:* The dual aim of this research was to consider the impact of providing the First Steps program on the stories of people with Parkinson’s Disease (PD) and to investigate the psychosocial and emotional mechanisms which may explain this impact. *Methods*: A qualitative study using a subtle realist paradigm and hermeneutic phenomenological methodology was undertaken. A single semi-structured interview was used to consider the impact and experiences of people with PD who completed either the intervention (2-day peer-led behavior intervention using storytelling 6–8 weeks apart) or received telephone support calls as part of the active control group. Descriptive statistics and a narrative analysis were undertaken on the results. *Results:* Forty-two participants were invited to participate, forty of whom completed the interview. This included 18 from the intervention group and 22 from the active control group. The intervention group identified the value of the program as worth-while, demonstrating improved exercise behavior and coping mechanisms following the intervention. Three major stories (the affirmed, the validated and the transformed story) identified the impact of the intervention. Three internal mechanisms (perceived control, hope and action, and the individual’s mind set) alongside three social mechanisms (social comparison, social control and the first opportunity to share with peers) appeared to explain this impact. *Conclusion*: This study provides exciting and novel evidence of the impact of a peer-led psycho-educational intervention for people newly diagnosed with PD. Further research is needed to consider the impact of stories-based approaches on participants and consider a critical evaluation of the mechanisms which may explain changes in stories and self-reported behaviour.

## 1. Introduction 

Parkinson’s is the second most prevalence neurodegenerative disease with a prevalence of the disease ranging between 108 and 250/100,000 (although it is higher for individuals over 65 years, e.g., up to 950/100000) for westernized countries [1]. In addition to movement impairments and motor symptoms, research has highlighted neuropsychiatric symptoms, such as mood, which subsequently can have a significant and detrimental impact on quality of life [2]. Two major psychosocial challenges exist following the diagnosis of Parkinson’s Disease (PD) [3]. These include a significant impact on social confidence, self-esteem and social competence as well as becoming self-conscious in social situations [3]. It also includes difficulty in psycho-emotional adaptation to the illness. However, it should be noted that people with PD can adopt psychosocial and behavioural strategies to overcome these factors [4]. 

A core treatment for PD following diagnosis is supportive therapies [5], which include the delivery of health care professional led treatment like physiotherapy [6]. However, the experience of diagnosis has consistently identified a lack of social support (e.g., perceived lack of value of information provided or ability to ask questions) in the care of people with PD at the time of their diagnosis [7,8,9]. As a response to this, education-based programs have been developed [10,11,12,13,14]. The benefits of these programs include self-reported mood and psychosocial improvements [13,14] as well as quality of life [1]. In contrast, two studies identified no improvement in quality of life [12,13]. More recently, research has used peer-led approaches and identified positive findings. For instance, peer-based interventions have been identified as improving behaviors like physical activity [15] and psychological well-being [16]. Further investigation into peer-led approaches would provide needed evidence on the benefits of rehabilitation programs for people with PD which is at presence scarce [3]. Further to this, review evidence [3] has highlighted that peer social support can positively impact psychosocial and behavioural well-being. 

The First Steps program is a peer-led program for people newly diagnosed with PD. The program is led by presenters who themselves have PD [17]. The First Steps program was derived from the European Parkinson’s Therapy Centre [18] with input from Parkinson’s UK Oxford Branch [19]. An important element of this program is the provision of storytelling by presenters with a focus on specific topics and through meeting peers. There is evidence of the value of using story-based educational interventions to (a) enhance research through patient participation [20] and (b) improve knowledge of health care professionals [21]. 

The First Steps program was able to consider if a peer-led Parkinson’s education group could change attitudes and behaviours through stories. This would be facilitated in two ways within the program: (1) Being able to explore and express one’s own story. For instance, past research has shown that just the ability to explore one’s own story can help people with PD renegotiate their social identity and cope [22]. (2) Research has also identified that telling and sharing stories can be used to influence attitudes and behaviours by ‘transporting’ the listener into the world of the teller [23,24]. One factor which may aid the effectiveness of the First Steps program could be the level of similarity the listener feels towards the story being shared by their peer [25]. The level of similarity and impact of the First Steps program can be documented by considering the master plot of the stories told by individuals following the intervention. The master plot illustrates how an individual psychologically adapts and copes when living with the illness. Research has identified 13 common plots (for instance, the Heroic master plot can illustrate how it is possible to overcome challenges, the Quest master plot can illustrate how living with an illness can be viewed differently) told by people with neurological illnesses [26]. The First Steps program allowed people with PD to listen to stories that illustrate adaptive behaviours and coping and contrast these stories with their own. This process provides access to and understanding of positive psycho-emotional adaptation [27,28]. It is important that research documents the impact hearing such stories by documenting the reactions and master plots of the listeners. It is also important that the mechanisms behind this process warrant further consideration. It is likely that particular psychosocial mechanisms are involved, given the importance of perceived similarity with others when stories are used to persuade others [25]. The mechanisms most likely behind any impact would include social comparison and/or social control as well as self-esteem [29]. Qualitative-based approaches would be well suited to understanding the ‘how’ and ‘why’ perceived changes occur. To the best of the authors knowledge this would be the first attempt to understand this.

Given the above, there is a need to understand if a peer-led psycho-educational intervention for people newly diagnosed with PD could impact their own story and how this was possible. Thus, the dual aim of this research was to consider the impact of providing the First Steps program on the stories of people with PD and investigate the psychosocial and emotional mechanisms which may explain this impact. 

## 2. Material and Methods

### 2.1. Study Design

A qualitative study based on hermeneutic phenomenology (study focusing on the interpretation of experience by participants) and situated within a subtle realist paradigmatic world view (a view which seeks to focus on common realities for participants and a view that a-priori knowledge can be used to enhance understanding of such realities) was undertaken. This qualitative study was assessed using a final interview to examine participant experiences of a feasibility study that used a step wedge pragmatic design. 

### 2.2. The Qualitative Researcher 

All semi-structured interviews were undertaken by researcher AS. He was a white male, aged 39 at the time of the interviews. He had 13 years of post-doctoral experience with qualitative research. No relationship with AS was gained prior to the phone interview. He was identified to participants by author (SL). Author SL informed participants of the aim of the process and identified a suitable time to call. Both authors were blind to the allocation of the participant until the time of the interview. 

### 2.3. Setting and Context 

#### 2.3.1. Control Group

The non-active control group received three phone calls from researcher (SL) and were based in Surrey (as the First Steps program was not yet available in this area). Each phone call contained standardised questionnaires assessing health and wellbeing, clinical service use, activities of daily living, physical activity and diet (as listed below). 

#### 2.3.2. Intervention Group

This section is presented according to TIDeR checklist [30]. 

*Name*: The First Steps program, UK.

*Why*: The program was co-designed by people with PD to enable support in coming to terms with a diagnosis of PD. A key element of the program is the opportunity to provide information around specific topics. This information is given by presenters who have PD and use experiences and stories to inform those who attend. Time is also allocated for participants to discuss and share stories. 

*What*: A two-day period is used to provide the following sessions. On day 1: (a) welcome (b) information regarding PD and medication, (c) how to face the future positively, (d) addressing fears and misconception, (e) accessing the right services, getting the right information and support, and (f) the importance of exercise and lifestyle in managing the condition. On day 2 (6–8 weeks later): (a) re-cap of day 1, (b) rights in relationship to PD including employment, driving, prescription, changing doctors, and further practical facts. 

Discussion activities: (a) review of participant’s physical activity and how participants have been getting on since day one, and (b) group discussions for loved ones on how they can best support their partner or relative. 

A single taster exercise session: the exercise session was provided in a group setting and was followed by the provision of information on local exercise classes for people affected by PD. 

*Who provided*: A Parkinson’s UK facilitator registered all people and scheduled all the sessions. Two people with PD led all the sessions, except the exercise session which was provided by a neurological physiotherapist (day 2 only). Other support staff were available to provide lunch and information. 

*How*: The delivery was face to face and arranged in groups. The group size of the intervention was aimed at around 5 people with PD and they could bring a partner, family member, friend, or carer each day of the intervention.

*Where*: The intervention took place in three locations (Oxfordshire, Hampshire or Devon) within non-clinical environments. Free parking, easy access to the location, a free lunch and access to exercise equipment was provided at each location. The intervention was not yet available in Surrey.

*When and how much*: The two-day program included specific information topics across both days (as outlined above). Both days had breaks for sustenance. Day 2 had a single group exercise session delivered within a hall setting. 

*Tailoring*: People with PD could ask questions on both days. 

*Modification*: No modification was made. 

Intervention Assessments

Both the intervention and control groups completed the standardised questionnaires over the telephone. The same questionnaires were completed at a 3 month and 6 months post baseline follow-up (after the first telephone assessment was completed). The participants’ spouse, partner, family member or friend (if consented) were also assessed for carer strain at each follow-up assessment. For the purpose of this study, only baseline data were reported to give an overview of participant demographics.

### 2.4. Procedures for the Qualitative Interview

All the participants were contacted using a single telephone interview conducted by author AS (following their 3 assessment calls conducted by SL). 

Participants were assessed for the feasibility study at baseline on enrolment into the trial. The first participant was enrolled onto the trial on 5 February 2018. For full details of the trial processes see Trial registration number ISRCTN14760402. Participants were reassessed at 3 and 6 months and as such the trial is currently ongoing. This research contains a sub-sample of participants selected from the trial until data saturation had been reached. 

The baseline demographics of the trial group as of 10 June 2019 were 53 participants (34 male, 19 female), with a mean age of 67.8 ± 7.8 years and time since diagnosis of 5.1 ± 3.2 months. The EQ5D mean was 79.9 ± 13.6. Full details of demographics will be available in a subsequent publication. 

As of 10 June 2019, 53 people had been recruited, of which 39 had reached the 6-month assessment point. Out of the 39 participants, 1 had withdrawn from the study prior to their 3 month assessment, leaving 38 participants with completed 6 month assessments. The reason for the withdrawal was not having enough time to complete the telephone calls. No participants were lost to follow-up. Out of these 38, two participants opted to complete the interview in paper-based forms by post due to difficulties hearing over the telephone. Therefore, 36 participants were available to interview over the telephone by researcher AS.

### 2.5. Sampling Strategy 

Purposive sampling of individuals who had taken part in the program was carried out. This was to represent a range of ages and to be split across the intervention and control group. Individuals were included if (a) they had been given a recent (within 12 months) diagnosis of PD, and (b) were above the age of 18 years. Individuals were excluded if (a) they had a clinical diagnosis of severe depression or psychosis, (b) they had reduced cognition that would preclude active involvement and capacity to consent to participate, or (c) they were unable to understand English.

### 2.6. Sample Size

The sample size was directed by the ability to critically consider the main themes and determined by data saturation [31]. Initially, 22 interviews were conducted and analysed. This generated a focus on the importance and impact of stories and storytelling and the psychosocial mechanism behind them. The final 18 interviews were focused on the content that followed to allow confidence in the saturation of sub-themes. 

### 2.7. Outcome Measures 

Demographics: The demographics recorded included age, gender, and time since diagnosis.

Primary outcome measure: A semi-structured interview to assess the experiences of the intervention, the feasibility and acceptability of the intervention, as well as the impact of the intervention in relation to the stories that were heard and shared. The interview schedule was pilot-tested. No changes were made to the original design. The interview schedule contained 2 closed questions and 16 open questions with several additional prompt questions. The Appendix A contains the interview schedule. 

### 2.8. Ethical Approval

Ethical approval was gained for the study from South Central—Hampshire A Research Ethics Committee on 23 August 2017 (17/SC/0346). Health Research Authority (HRA) approval was gained on 8 September 2017.

### 2.9. Analysis

Descriptive statistics were used to analyse age, time undertaken since the start of the trial and time since diagnosis. A narrative analysis focusing on master-plots [27] (common stories plots told by participants) was used to explore and explain the impact of the program on individuals with PD. A thematic analysis using a-priori concepts was used to consider the established psychosocial mechanisms of impact [29] on the stories of individuals. Two general stages of analysis were undertaken: Stage 1 included open coding and immersion in interviews and multiple reading of scripts. Stage 2 identified a mind map of the most common comments or themes relating to the psychosocial mechanisms of physical activity or behavioral impact and or change. Stages 3 to 6 sought to identify common story plots that illustrated impact from the First Steps program and identify the likely psychosocial mechanisms behind the impact. The final stage is available in the Appendix A with verbatim quotes. An audit trail is available on request from the AS. No computer program assisted the analysis. 

### 2.10. Trustworthiness

Trustworthiness was considered by the use of reflexivity by the researcher by adhering to the COREQ guideline [32] and by documenting the follow strategies to enhance rigor [33]: (a) an audit trail, (b) attention to negative cases (searching for instances when participants directly contrasted or contradicted the themes and content identified) within the analysis, (c) peer debriefing (AS presented a defendable case to other authors HD, SL, JC) for critical comment after the first 22 interviews, and (d) theoretical triangulation, as themes were fitted with existing knowledge of psychosocial mechanisms which impact physical and mental health [29]. These steps are in-line with studies that situate themselves within a subtle-realist paradigm [3]. Further to this, AS was required to provide a defendable case to the research group about the findings. The presentation of the findings to the group was made after interview 24. Following this, interviews were focused on exploring the themes and saturating content identified. Quotes are presented verbatim and identified by a participant number, e.g., P1M52 (Participant number: P1; gender: male; age: 52 years old).

## 3. Results 

### 3.1. Demographics 

Forty-two individuals (16 female, 26 male) were phoned. One male (P31M74) did not respond to phone calls or follow up postal questionnaires. One male (P28M74) declined participation at the point of the phone call with AS. He identified that the data collection process had taken up too much time and he could not give any further time to the study. Across the remaining 40 participants, the average age was 66.7 ± 9.0 years (control 68.4 ± 10.7 years; intervention 66.2 ± 7.5 years). The average time since diagnosis at the end of the interviewing process was 13.5 ± 4.6 months (control 13.2 ± 4.7 months; intervention 14.7 ± 4.8 months). The average time since diagnosis at the point of enrollment into the study was 5.4 ± 3.3 months (control 5.2 ± 3.4 months; intervention 5.8 ± 3.4 months). Eighteen individuals (18/40, 45.0%) received the intervention, 22 (22/40, 55%) received the active control condition. Appendix A provides a full break down of the demographics. 

### 3.2. Non-active Control Group Responses

All the individuals in the active control group perceived that the intervention had no impact or resulted in no change. No individual in the control group had contact with the intervention group. When asked about the value of participating in the active control group, the following statements were made: (a) that there was no value (P47M60) or that it was difficult to determine (5/20; 25.0%; P9M47; P23F54, P21M73; P51F60; P74M70), (b) that talking to someone had value (8/20; 40.0%; P9M47; P10M63; P20M72; P34F74; P43M60; P48M80; P54F56; P64M71). For instance, P48M80 stated “*it is interesting to know that people are interested in how things are going. But umm, I wasn’t expecting it to make any difference to my actual condition*”. This could be seen in a more critical way for participant FS20 who stated: “*outside of talking to you [AS] and the GP, if I ever get to see them, I don’t talk to anybody*.”, and (c) it was identified that there was some impact in thinking about aspects related to behaviour. One individual (P5M75) said that it got them “*thinking about aspects like sitting”*. 

### 3.3. Intervention Group Responses—Closed Answer Questions

All the individuals identified the program as (a) very worthwhile to be a part of, (b) being undertaken at a good and appropriate location and (c) having staff that were competent. Six individuals (6/18, 33%, P5M75, P14M69, P18F57, P19M55, P24M76, P26M83, P35F59, P36F58, P38M63) identified that they were ‘feeling better’ or ‘more positive’. The most consistent impact was identified as changes in attitudes or behaviours of participants. 

### 3.4. Intervention Group Responses—Synthesis of Qualitative Responses About the Course 

**Theme 1:** The response and impact on the stories of participants 

The first theme considers the impact of the First Steps Program according to the stories shared during the interview. This theme had three sub-themes: (a) the affirmed story (no behaviour change), (b) the validated story (where individuals were affirmed and understood how to engage in physical activity) and (c) the transformed story (where behaviour change was created as a direct result of the First Steps program). 

**Sub-theme 1a:** The affirmed story

The affirmed story was told by individuals who identified that the program provided no extra knowledge for them. It was defined by an acknowledgement that their past choices and behaviours were affirmed by the First Steps program and that this knowledge remained unchanged. This implied that physical activity levels were unchanged. Five individuals (5/18; 28%; P5M75, P6M64, P16M69, P17F70, P26M83) identified that the First Steps program affirmed their choice but led to no change in behaviour. For instance, P5M75 stated: “*I mean it [physical activity] is not something that you need to stop doing.*” For P16M69 and P17F70 the Parkinson’s had not progressed, so no lifestyle changes were needed. Related to this, P26M83 stated: “*as much as possible I forget about Parkinson’s Disease [and carry on as before]*”. For P6M64, the same barriers to exercises existed after the course and no changes were perceived possible. 

**Sub-theme 1b:** The validated story

The validated story identified that the First Steps program had helped them understand and validate the importance of using the information gained. The individuals had received the stories and experiences of others and planned to adopt similar behaviors. It was evidence of the presenter’s stories becoming their own story. They appeared more motivated to continue physical activity or exercise. The course had provided a reinforcement of the message that physical activity had to be part of their lives with Parkinson’s. Terms used by the presented were remembered and repeated during the interview, for instance, the need to undertake ‘strong’ exercise or increase what they were doing. The validated story was identified in 7 individuals (7/18; 39%; P2M68, P16M69, P8F70, P24M76, P26M83, P32M60, P36F58). For instance, for P36F58, it validated the choices made: “*I think it validated what I felt I should be doing…For instance, if I don’t do that [exercise], I won’t be able to do the other stuff anyway. It is about knowing you’re not being selfish when you look after yourself*”. Alternatively, P8F70 identified the reinforcement of the positive benefits that physical activity could bring by stating “*I think I have always felt better when I have done exercise classes*”. This story could also include statements that enabled greater engagement with physical activity. Individuals identified small changes learnt from the course that allowed this, for instance, knowing their limits regarding physical activity participation (P2M68) and having a better understanding of the choice and technique of specific exercises (P26M83). 

**Sub-theme 1c:** The transformed story

The final story, the transformed story, identified the transformation of behaviour and a subsequent change of physical activity or exercise. The change was attributed to the course and related to an increase in the level, intensity or type of physical activity or exercise engaged in. This was directly attributed to the First Steps Course. This was identified by 8 participants (8/18; 44%; P1F61, P2M83, P5M75, P8F70, P14M69, P18F57, P19M55, P35F59). The changes identified included beginning a new class like Zumba Gold or Pilates (P1F61, P2M68, P5M75, P19M55), joining a gym (P18F57), doing more walking or walking better (3/18; 17%; P1F61, P14M69, P19M55) and changing a fitness regime or doing more in general (4/18; 22%; P8F70, P18F57, P26M83, P35F59).

**Theme 2:** Psychosocial Mechanisms explaining the benefits from receiving stories 

Many participants identified psychological and social reasons or mechanisms which explained the impact of the First Steps program. These were identified with seven sub-themes. This included three internal sub-themes: (a) perceived control, hope and action, (b) the individual’s mind-set, and (c) perceived confidence. It also included three social sub-themes: (d) social comparison, (e) social control, and (f) the first opportunity to share with peers. 

**Sub-theme 2a:** Perceived control, hope and action

This sub-theme was generated as a result of participants seeing that a future was possible by listening to the stories of the presenters and others. Individuals perceived that action could be taken as illustrated by others who are sharing their own experience. This theme was supported by nine participants (9/18; 50%; P1F61, P2M68, P8F70, P14M69, P32M60, P35F59, P36F58 and P38M63). The theme was illustrated by the recognition of the importance of physical activity and exercise in maintaining physical well-being. For instance, P32M60 stated: “*if you as a Parkinson’s patient want to keep active, want to keep your mobility, balance and things of that nature, you have to put in the effort and do the exercise*.” P14M69 summarised the theme and the importance of taking control and action to enable living with PD: “*AS: in terms of getting on with your life? What impact would it [the First Steps programme] have had on that? Any change? P14M69: well, I, I think, what was important for me at that stage was that you could actually, fight back to some degree. Take the future in your own hand to some degree and that is quite important*.” This theme was a key illustration of the perceived impact of the stories shared by participants and the outcome of social comparisons and the ability to relate to others and take the message onboard to utilize it in their own life situation. 

**Sub-theme 2b:** The individual’s mindset

Many individuals talked about the impact on mood and mindset from the program, identifying that the First Steps program provided a protective mindset against the experiences of the symptoms and an acceptance of the feelings or emotions about having PD. For instance, P18F57 stated “*I don’t want to just stagnate. Whereas before I was sort of like, feeling sorry for myself I suppose. I have a positive attitude*.[now]”. This theme was identified by 11 individuals (11/18; 61%; P1F61, P2M68, P6M64, P13M84, P14M69, P18F57, P19M55, P24M76, P32M60, P36F58; P38M63). The impact on an individual’s mindset had several influences. This included thoughts about the program before attending because of the opportunity to give and share with others (P2M68), the value of meeting varied people with different outlooks (P38M63). Furthermore, individuals highlighted a positive atmosphere within the group (2/18; 11%; P1F61, P18F57). This was identified as important because of the worry (P24M76), difficulty (P19M55) and sadness (P14M69) that was associated following the diagnosis of PD. P13M84 noted that the content of the session could be depressing. P13M84 only attended the first session and it was noted by another participant that the experience of the second session could be more positive: “*people [who were] negative on first sessions [were] more positive on next session*” (P6M64). This theme linked to hope because it related the impact the course had on psychological adaptation and emotions. 

**Sub-theme 2c:** Perceived confidence 

An increase in self-efficacy and being confident about living with PD and illustrated by how others achieve that was identified by 6 individuals (6/18; 33%; P1F61, P2M68, P14M69, P18F57, P36F58, P37M71). It was identified through stories of how to engage in living with PD. For instance, this could include the confidence to identify that you have PD in social situations. P2M68 stated: “*one of the presenters made it quite clear that whatever stage you are at, you should never be afraid to put your hand up and say I can’t do this because I have Parkinson’s*”. Second to this, participants identified the confidence to do more activities and interaction. Examples of this include the confidence to enroll in a course and join the gym (FS18), the confidence to attend a dinner party without worry about wreaking other people’s night (FS14). This theme could have been related to the successful engagement in physical activity following the program which had a direct influence on individual self-efficacy. 

**Sub-theme 2d:** Social comparison 

An essential mechanism explaining the impact identified above was through social comparison. Most frequently, this was identified as being able to relate to another individual who has had a similar experience. Relatedness was identified through listening and sharing stories that demonstrated a positive and possible way to live with PD. Within each day, there was also an opportunity to talk with others, listen and understand others’ stories. This theme was supported by 14 individuals (14/18/ 78%; P1F61, P2M68, P5M75, P7F63, P13M84, P14M69, P18F57, P19M55, P24M76, P26M83, P32M60, P35F59, P36F58, P38M63). Most comments related to the benefit of understanding others’ experience and broadening one’s own knowledge of PD. For instance, P17F70 stated: *“it is interesting for me to see other people, you know how different we all are*”. This process primarily allowed individuals an insight to how others coped and managed the condition and a chance understanding how others viewed or let the PD affect them. This theme was captured by P2M68, who summarised the value by stating: 


*“it was from listening to the presenters who have had Parkinson’s for such a long time, umm, and how, how severe they have got it and how well they have coped with it and their coping mechanisms, made me realise, I am at the beginning of the journey that they are already on, umm, and they are obviously very wise because they have learnt a lot about their disease and management and umm, you know all sorts of coping strategies, both psychological and physical and I need to listen and take them on-board.”*


There were two instances (2/18; 11%; P26M83, P18F57) where this was identified as not so beneficial; P26M83 identified that the group appeared “*concerned about giving away their own position*”. These concerns illustrate that the benefits of sharing were not always occurring. Alternatively, another participant (P18F57) noted that one lady was reluctant to embrace ideas from the group.

**Sub-theme 2e:** Social control

Social control was identified within the mechanisms as participants could identify direct instances of when advice was given to them and taken on-board and embraced. This was identified by nine individuals (9/18; 50%; P1F61, P2M68, P5M75, P7F63, P8F70, P16M69, P26M83, P35F59, P36F58). It was reflected by general statements regarding physical activity and exercise and undertaking more intense exercise and devoting more time to it. It also was reflected in specific comments that illustrated a change in physical activity or exercise like participating in a new Pilates or yoga class. For instance, P2M68 stated: “*one lady suggested I went out and tried Zumba gold which was very helpful*”. This then went further than just including physical activity as advice was also given about living with PD and having access to aspects which improve life like a radar key, or when to take medication. Some of this was summed up generally, for instance, P7F63 stated that the benefit was through “*meeting people with the same condition and comparing notes, it is interesting*”. The benefit of this understanding was also identified for people’s partners. This was mentioned by three individuals (3/18; P16M69; P18F57; P19M55). 

**Sub-theme 2f:** the first opportunity to share with peers 

The First Steps program was identified as an essential opportunity to meet with others and share stories and experiences. Six people (6/18; 33%; P1F61, P2M68, P5M75, P32M60, P35F59, P38M63) identified that they had not met others with PD before the course. This group identified that they would not have known the valuable information if it had not been the course. P6M64 stated that it would be beneficial to receive this information closer to diagnosis (he was enrolled onto First Steps 6 months after diagnosis). Importantly, one individual (P2M68) identified that she would not do anything like the course again. Furthermore, P8F70 was very complimentary of the First Steps program and valued listening to peers, but did not want to share with others. She stated: 


*“well, I made a conscious decision not to talk [about PD], other than sharing it with some immediate family members. I made a conscious decision not to say anything, on the wider side of things, because I know, of two or three people who had diagnosis of Parkinson’s and also the reactions that people have given and my feeling, for me personally, was that, until the time comes when it is obvious enough that people need to know, I don’t need to share that”*


## 4. Discussion 

This study illustrates the behavioral impact of the First Steps program and the importance and impact of peer stories. Our results are able to further the scarce evidence and show that people can value, remember, and act to change their lifestyle and that this change was likely due to the relatedness they felt towards the stories from peers. This study supports past psycho-educational interventions regarding the positive benefits experienced by people with PD and extends these findings to people who are newly diagnosed. The identified benefits included perceived change in attitude and behaviours following the intervention as well as benefits related to psycho-emotional adaptation and coping. 

The critical mechanisms that appeared to influenced participants stories included social comparison, relatedness, and motivation to adopt the view of the presenters and others in the group. Enhancement of the individual’s physical activity and exercise social identity was likely across participants as they were often able to relate to the presenters and peers within the First Steps program. This process could be similar to a process described as reframing the social identity following participation within an exercise program [34]. This process likely leads to positive psycho-emotional changes as the impact of peer stories impacts their own. This supports previous theoretical evidence regarding this process [27].

Storytelling is regarded as an effective way to enhance education and learning for individuals with chronic disease [35]. Review evidence has identified a lack of studies that use story telling for people with PD [36]. In other populations, storytelling interventions have been associated with encouraging positive attitudes and behaviors, such as physical activity and dietary changes [28,37]. This current study supports these findings for people with PD. The adoption of similar stories meant that following the First Steps intervention, participants’ past experiences and stories can change and evolve from potentially negative stories to more positive ones.

Exercise has been recognized as a possible adjunct therapy for individuals newly diagnosed with PD which has the potential to reduce disease progression and aid motor symptoms [4,38] as well as promote global cognitive function [39]. Within the current study, it is possible that physical activity and exercise behaviors were enhanced by sharing information about one’s individual ability to cope and live with the disease [40]. The current results provide an illustration that the poor experience of communication of disease and its impact at diagnosis [7] can be buffered and supported by a peer educational storytelling intervention. 

The perceived impact on positive behaviours supports past work on the value of peers within psychoeducational interventions [15] and illustrates the importance of people with PD knowing how to engage in activities and behaviors [41]. It is likely that the findings identified here are in part the result of individuals adopting positive coping strategies to manage PD [42]. The reasons for this likely include an impact of the program on the individual’s mind-set, confidence, and acceptance of a condition through relatedness with others. These findings support the findings by Hellqvist et al. [16]. It is also important to note that the participants who reported no change in attitude still identified the event as being worth it. The main reason for this was due to being with peers, which likely had value, e.g., reduced fear, feeling accepted, considering the possibility of different attitudes and behaviours and affirming past choices. This is important because in the absence of reporting behaviour change, participants still valued the program. Where no behavior change was reported, the benefit appeared to be on a participant’s psycho-emotional well-being. This has been identified in other studies, e.g., [28].

The shared experience of the presenters likely represented a positive psycho-emotional response to adapting to the diagnosis. This may occur through key mechanisms [27]: (a) an ability to aid acknowledgement of present constraints and the impact of PD and (b) an ability to enforce an idea of possibility about what changes may occur in the future, how these changes can be managed, and an understanding of the emotional impact of PD though the observation of positive outcomes and encouraging stories from peers. 

### 4.1. Limitations 

Several limitations are acknowledged: (a) The analysis focused on the impact of considering attitudes and behaviours most frequently related to physical activity and exercise. This may have limited the focus of the current article. (b) The approach may be limited to a self-selected sample. (c) The culture, environment, level of education of participants and setting may be, to some extent unique. (d) The geographical location of the First Steps program may prevent some people attending. This could limit the representativeness of the sample. (e) This study should not be considered as an exact documentation of behaviour change. The results must be viewed as common realties experienced by participants. 

### 4.2. Implications 

Specific implications were identified: (a) The First Steps program could represent a very good supplement to support individuals newly diagnosed with PD. It is likely that the program can support or enhance their attitude and behaviour towards physical activity and exercise. (b) Sharing experiences related to physical activity and coping behaviour likely impacts the bio-psycho-emotional and social well-being of newly diagnosed individuals with PD. (c) Social comparison and social control are evident as two mechanisms which appeared to aid changes in perceptions of behaviour and physical activity as a direct result of hearing, sharing and comparing stories. However, benefits often existed even if change did not occur as participants valued the connection and relatedness of experiences to peers. This finding should be considered more widely within rehabilitation programs. (d) Psycho-emotional adaptation and hope likely play a central role in allowing perceived changes in attitude and physical activity behaviour to occur. A patient co-delivered workshop appeared to allow this by being immersed in a positive group environment associated with positive views, perceptions and experiences about managing PD. This exchange and sharing could allow seeing possibilities and could help others move past a simple acknowledgement of limitations caused by symptoms to embrace limitations. Considering the change in psycho-emotional adaptation is essential to establish this in further research. 

## 5. Conclusions

This study provides novel evidence for the impact of a peer-led psycho-educational intervention for people newly diagnosed with PD. In particular, this study supports the notion that positive stories from peers can support and encourage positive attitudes towards behavioural change. The storytelling element of the First Steps program could be used as an intervention strategy that can be explored in other illness conditions and other geographical locations. However, further research is needed to consider the impact of stories-based approaches on participants and consider a critical evaluation of the mechanisms which may explain changes in stories and self-reported behaviour.

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
