# Peer review of "A Qualitative Study on the Impact of First Steps—A Peer-led Educational Intervention for People Newly Diagnosed with Parkinson’s Disease"

_behavsci, 2019, doi:10.3390/bs9100107_

Round 1

Reviewer 1 Report

Thank you for the opportunity to review this manuscript. 

I appreciate if the authors will underline the novelty of their research related to previous researches and findings. 

The introduction need to be extend to highlight the actual information  related to the article topic. 

I recommend to add 2-3 more relevant conclusions according with your research findings. 

Author Response

Reviewer 1 comments

R1: Thank you for the opportunity to review this manuscript. I appreciate if the authors will underline the novelty of their research related to previous researches and findings. 

AS et al: Thank you for your responses and comments. We have now expanded on the novelty of this research in relation to previous studies in the introduction.

R2: The introduction need to be extend to highlight the actual information related to the article topic. 

AS et al: Thank you for this comment. We looked at the results carefully to identify required information. We have updated the information to lead the reader into the topic area and further rationale.

R2: I recommend to add 2-3 more relevant conclusions according with your research findings. 

AS et al: We have provided two more conclusions at the end of the manuscript.

Reviewer 2 Report

The manuscript entitled "A qualitative study on the impact of First Steps- a peer-led, educational intervention for newly diagnosed individuals with Parkinson’s Disease" is an innovative study about psycho-educational intervention specially for people newly diagnosed with Parkinson’s. Limitations are there, but they have discussed that in their manuscript, including self-selection of samples, and their culture, environment, level of education etc. This can be over come if this kind of study approach will be taken by other research groups in different geographical areas.

Author Response

Reviewer 2 comments

The manuscript entitled "A qualitative study on the impact of First Steps- a peer-led, educational intervention for newly diagnosed individuals with Parkinson’s Disease" is an innovative study about psycho-educational intervention specially for people newly diagnosed with Parkinson’s. Limitations are there, but they have discussed that in their manuscript, including self-selection of samples, and their culture, environment, level of education etc. This can be over come if this kind of study approach will be taken by other research groups in different geographical areas.

AS et al: Thank you for these comments. We have used this advice to develop our conclusions in the Discussion section.

Reviewer 3 Report

The study from Soundy A and colleagues deal with the impact of First-step- a Peer-led, Educational Intervention for Newly Diagnosed Individuals with Parkinson's Disease. The topic is interesting as it might confer knowledge about changes in Parkinson's Disease.

Therefore, the manuscript meets our criteria, and the only improvements that should be made are to the language.

Minor points:

As an example, the title:  "Diagnose Iindividuals" should be changed for Individuals.

Parkinson's for Parkinson's Disease and adds the acronym (PD).

Among others

Author Response

Reviewer 3 comment

The study from Soundy A and colleagues deal with the impact of First-step- a Peer-led, Educational Intervention for Newly Diagnosed Individuals with Parkinson's Disease. The topic is interesting as it might confer knowledge about changes in Parkinson's Disease.

Therefore, the manuscript meets our criteria, and the only improvements that should be made are to the language.

AS et al: Thank you for these comments. 

Minor points:

As an example, the title:  "Diagnose Iindividuals" should be changed for Individuals.

AS et al: We have change this as requested. 

Parkinson's for Parkinson's Disease and adds the acronym (PD).

AS et al: We have adopted this acronym.